# CD147/Basigin Is Involved in the Development of Malignant Tumors and T-Cell-Mediated Immunological Disorders via Regulation of Glycolysis

**DOI:** 10.3390/ijms242417344

**Published:** 2023-12-11

**Authors:** Takuro Kanekura

**Affiliations:** Department of Dermatology, Kagoshima University Graduate School of Medical and Dental Sciences, 8-35-1 Sakuragaoka, Kagoshima 890-8520, Japan; takurok@m2.kufm.kagoshima-u.ac.jp

**Keywords:** CD147, glycolysis, malignant tumor, psoriasis, immune disorder

## Abstract

CD147/Basigin, a transmembrane glycoprotein belonging to the immunoglobulin superfamily, is a multifunctional molecule with various binding partners. CD147 binds to monocarboxylate transporters (MCTs) and supports their expression on plasma membranes. MTC-1 and MCT-4 export the lactic acid that is converted from pyruvate in glycolysis to maintain the intracellular pH level and a stable metabolic state. Under physiological conditions, cellular energy production is induced by mitochondrial oxidative phosphorylation. Glycolysis usually occurs under anaerobic conditions, whereas cancer cells depend on glycolysis under aerobic conditions. T cells also require glycolysis for differentiation, proliferation, and activation. Human malignant melanoma cells expressed higher levels of MCT-1 and MCT-4, co-localized with CD147 on the plasma membrane, and showed an increased glycolysis rate compared to normal human melanocytes. CD147 silencing by siRNA abrogated MCT-1 and MCT-4 membrane expression and disrupted glycolysis, inhibiting cancer cell activity. Furthermore, CD147 is involved in psoriasis. MCT-1 was absent on CD4^+^ T cells in CD147-deficient mice. The naïve CD4^+^ T cells from CD147-deficient mice exhibited a low capacity to differentiate into Th17 cells. Imiquimod-induced skin inflammation was significantly milder in the CD147-deficient mice than in the wild-type mice. Overall, CD147/Basigin is involved in the development of malignant tumors and T-cell-mediated immunological disorders via glycolysis regulation.

## 1. Introduction

CD147/Basigin, originally cloned as a carrier of the Lewis X carbohydrate antigen, is a transmembrane protein belonging to the immunoglobulin superfamily [1,2]. CD147 performs pleiotropic functions by binding to various molecules, including monocarboxylate transporters (MCTs). MCTs, namely, MCT-1 and MCT-4, play important roles in the regulation of glycolysis, the enzymatic conversion of glucose to pyruvate and adenosine triphosphate (ATP), to fuel the pathophysiological activities of cellular growth and proliferation [3]. CD147 is highly associated with MCT-1 and MCT-4 and facilitates their expression on the plasma membrane [4].

Malignant melanoma (MM) is highly malignant because of its rapid growth and metastatic properties. In MM, CD147, whose expression is increased on the cell membrane, plays an important role in tumor proliferation, the production of matrix metalloproteinases (MMPs), angiogenesis, invasiveness, and metastatic activity in vitro and in vivo [5,6]. In human MM cells, CD147 co-localizes with MCT-1 and MCT-4 on the plasma membrane and promotes the proliferation, invasiveness, and metastasis of MM by regulating glycolysis. The silencing of CD147 by a small interfering RNA (siRNA) clearly abrogates the expression of MCT-1 and MCT-4 and their co-localization with CD147 and dramatically decreases the glycolysis rate, extracellular pH, and the production of ATP. Thus, cell proliferation, invasiveness, and VEGF production are significantly decreased by siRNA. These previous observations strongly suggest that CD147 interacts with MCT-1 and MCT-4 to promote tumor cell glycolysis leading to the progression of MM [5,6,7].

T cells also depend on glycolysis to generate energy for their differentiation, proliferation, and activation [8]. Psoriasis is a chronic inflammatory skin disease with a persistent episode of clearly defined scaly erythematous patches. The development of psoriasis depends on the differentiation of naïve CD4^+^ T cells into Th17 cells [9,10]. We hypothesized that CD147 plays a role in the development of psoriasis lesions through the regulation of glycolysis and demonstrated that CD147 is crucial for the development of psoriasis through the induction of Th17 cell differentiation. Serum CD147 levels were increased in psoriasis patients and the CD147 and MCT-1 expressions were elevated on their CD4^+^ RORγt^+^ Th17 cells in the dermis. The ability of naïve CD4^+^ T cells to differentiate into Th17 cells is absent in T cells from CD147-deficient mice in vitro. In CD147-deficient mice and mice lacking CD147 in their bone marrow hematopoietic cells, imiquimod (IMQ)-induced psoriasis-like dermatitis is remarkably mild. These results show that CD147 is necessary for the development of psoriasis through the promotion of Th17 cell differentiation [11].

Here, we review the function of CD147 in glycolysis and its involvement in the development of malignant tumors and immune disorders.

## 2. CD147/Basigin

### 2.1. Discovery of CD147/Basigin

Cell surface glycoproteins regulate cellular activities such as proliferation, differentiation, adhesion, migration, and transmembrane transportation [12,13]. CD147/Basigin was cloned as a carrier of Lewis X, a cell surface carbohydrate antigen expressed in embryonal carcinoma (EC) cells using the λgt11 expression library of F9 EC cells in our study of developmentally regulated cell surface markers [1]. CD147/Basigin is a transmembrane glycoprotein comprising two immunoglobulin (Ig)-like extracellular domains, a single transmembrane domain, and a short C-terminal cytoplasmic tail [1]. Extracellular Ig-like domains have strong homology with the Ig variable domain and the major histocompatibility complex class II β-chain. Based on the structural analysis, CD147/Basigin was identified as a new member of the Ig superfamily [1]. CD147/Basigin is expressed at similar levels in various adult organs, mouse embryos, and EC cells. Because of its broad distribution, we considered that this molecule has basic or fundamental functions and termed “Basigin (Bsg)” as an abbreviation of basic immunoglobulin [1]. It is located on chromosome 19 at p13.3 and consists of 10 exons spanning approximately 12 kb [14]. The molecular weight of this protein portion is 28 kDa. CD147 has three asparagine (ASN) glycosylation sites on its Ig-like domains and is highly glycosylated. The molecular weights of the glycosylated forms ranged from 43 to 66 kDa. Their diverse molecular weights are due to the different modes of the glycosylation of molecules from various organs [2].

### 2.2. Discovery of Related Molecules

Identical molecules have been found independently from various perspectives in several laboratories and are given different names: M6, extracellular matrix metalloproteinase inducer (EMMPRIN), HT7, neurothelin, 5A11, gp42, OX-47, and CE9 [15,16,17,18,19,20,21,22]. Among these, M6 was cloned using peripheral granulocytes from patients with rheumatoid arthritis (RA) and was identified as a human leukocyte activation antigen [15]. EMMPRIN, previously known as tumor cell collagenase stimulatory factor (TCSF) [23], is implicated in the induction of matrix metalloproteinases (MMPs). Its expression is enhanced in various human carcinomas, including MM, and correlates with tumor progression and invasion by inducing the production of MMPs by stromal fibroblasts [24,25,26,27,28,29]. HT7 is a highly glycosylated protein localized in brain endothelial cells that is a receptor involved in cell surface recognition at the blood–brain barrier (BBB) [17]. Neurothelin is an inducible cell surface glycoprotein of BBB-specific endothelial cells and distinct neurons and regulates the interaction between vascularization and neuronal differentiation [18]. Fadool and Linser reported that 5A11 is a functional molecule involved in neural–glial mutual recognition in the avian neural retina [19]. A murine fibroblast membrane glycoprotein, gp42, is a fibronectin-receptor-associated antigen involved in neural cell adhesion [20]. OX-47 is a lymphocyte activation antigen. It presents in lymphocytes whose expression is markedly induced on activation with mitogens [21]. CE9 is a posterior-tail domain-specific integral plasma membrane glycoprotein cloned from a rat spermatozoon and plays an important role in spermatogenesis [22]. These studies suggest that CD147/Basigin is a multifunctional molecule involved in various physiological and pathological phenomena [30].

The gene and protein names given to these molecules are Basigin both in humans (Locus Link) and mice (Mouse Genome Informatics) and the symbol provided by the Human Genome Organization is *BSG* in humans [14] and *Bsg* in mice [31]. Because Basigin is expressed in leukocytes, the Cluster of Differentiation Nomenclature “CD147” was given at the 5th International Workshop in 1993 [32].

## 3. Chaperone-Like Function of CD147/Basigin

Previous studies have revealed that CD147 has various binding partners such as cyclophilin A (CyPA) [33], integrins [34,35], P-glycoprotein [12,30,36], and MCTs [4]. The pleiotropic functions of CD147 are attributable to its binding partners. CD147 acts as a chaperone for their proper plasma membrane expression and catalytic activity and participates in many pathophysiological processes through these molecules [12,30].

CyPA, the major target of the immunosuppressive drug cyclosporin A, is a ubiquitously distributed intracellular protein. CyPA is secreted by cells in response to inflammatory stimuli and is a potent neutrophil and eosinophil chemoattractant. In the inflammation process, CD147 acts as a cell surface receptor for CyPA and initiates signaling cascades leading to ERK activation [33]. CD147 binds to β1-integrin, which, in turn, induces the aggregation of promonocyte line U937 cells via protein tyrosine phosphorylation. The antibody against CD147 inhibits the aggregation and the tyrosine phosphorylation by blocking the binding between β1-integrin and CD147 [34]. The multidrug resistance (MDR) of cancer cells is often associated with the overexpression of P-glycoprotein (P-gp), a transmembrane ATP-dependent transporter. The expression of P-gp and CD147 is upregulated in the adriamycin (ADR)-resistant human mammary carcinoma cell line MCF7 (MCF7/Adr) compared to its non-ADR-resistant counterpart MCF7. The silencing of CD147 in MCF7/Adr by siRNA targeting CD147 resulted in the downregulation of P-gp expression and a reduction in drug resistance [36].

The association between CD147 and MCTs yielded important clues for understanding the role of CD147. Cross-linking studies showed that CD147 forms a homo-oligomer [37]. Homodimerized CD147 binds to two MCT monomers and facilitates its proper folding and expression on the cell membrane (Figure 1). In MCT-transfected COS cells, expressed MCT proteins accumulate in the perinuclear compartment, whereas co-transfection with CD147 cDNA enables the expression of functional MCT-1 or MCT-4 on the plasma membrane [4]. As mentioned previously, CD147 consists of two Ig-like extracellular domains: a single transmembrane domain and a short cytoplasmic tail at the C-terminus [1]. MCT-1 attaches itself directly to the transmembrane domain of CD147 [30,38]. The cryo-EM structure of the CD147/MCT complex has been determined [39,40] and Glu218 in the CD147 transmembrane domain is the binding site for MCT-1 [39,41]. Philp et al. demonstrated the role of CD147 in the retina, in association with MCTs [42]. CD147-deficient mice exhibit visual impairment. Their morphology of the retina at the light microscopic level and the fundus and the fluorescein fundus angiography appeared to be normal until 8 weeks of age, whereas the amplitude of all components of both the photopic and scotopic electroretinograms was decreased, indicating that both rod and cone functions were severely affected [43]. In these mice, the cell membrane expression of MCT-1 and MCT-4 was greatly reduced in retinal photoreceptor cells and adjacent Müller cells. Müller cells are retinal glial cells, whose major role is to maintain the functional and structural stability of photoreceptor cells. Müller cells provide photoreceptor cells with lactate as fuel for normal functions [42]. Because of the absence of MCTs on the membrane of facing cells, the flux of lactate from Müller cells to photoreceptor cells is disrupted, and photoreceptor cell activity is lost owing to energy depletion [42]. The absence of CD147 results in the impaired expression of MCTs on the cell membrane, leading to loss of vision.

Other binding partners of CD147 include glucose transporter-1 (GLUT1), CD44, the major hyaluronan receptor, CD43, CD98, γ-secretase, NOD2, γ-catenin, platelet glycoprotein VI (GPVI), and apolipoprotein D. Molecular interactions between CD147 and these binding partners have been previously documented [30].

## 4. CD147 and Glycolysis

The elucidation of the mechanism of lactate flux by CD147 and MCTs in the retina prompted us to investigate the role of CD147 in glycolysis. Glycolysis is the enzymatic conversion of glucose to pyruvate to generate energy, which is stored in the form of ATP. Pyruvate is further converted to lactic acid, which is exported through the plasma membrane and is required for metabolism and intracellular pH regulation [38]. Lactic acid is transported by proton-linked/lactate co-transporters, MCTs, on the plasma membrane. The detailed metabolic pathway of glycolysis was initially studied in cancer cells. Under physiological conditions, cellular energy is provided by mitochondrial oxidative phosphorylation, and glycolysis results from anaerobic enzymatic conversion. Warburg first reported that cancer cells depend on glycolysis for energy even in the presence of oxygen, i.e., under aerobic conditions [3]. They take up excess glucose through GLUT-1 or GLUT-3, which is then enzymatically metabolized to ATP to drive the pathological processes involved in cell division and growth [44]. In cancer cells, pyruvate is transformed into lactic acid during aerobic glycolysis and then released from the cytoplasm into the extracellular milieu by means of MCTs. Altered metabolism requires tumor cells to rapidly efflux lactate into the surrounding microenvironment to prevent self-poisoning. MCTs facilitate proton-linked monocarboxylate transport, leading to a decrease in the extracellular pH of tumors. The acidity of the tumor microenvironment produces more aggressive phenotypes in cancer cells that exhibit increased proliferation, invasiveness, metastasis, and VEGF production [45,46,47,48].

## 5. CD147 in Cancer Cell Glycolysis

Based on the observation in the retina, we investigated the involvement of CD147 in cancer cell glycolysis using MM cells. Human MM cells (A375) expressed higher levels of MCT-1, MCT-4, and CD147 and showed an increased glycolysis rate compared to normal human melanocytes. CD147 co-localized with MCT-1 and MCT-4 on the A375 cell membrane. CD147 silencing by siRNA abrogated MCT1 and MCT-4 expression and their co-localization with CD147. The glycolysis rate and ATP production were dramatically decreased and the extracellular pH increased. Subsequently, cell proliferation, invasiveness, and VEGF production were significantly inhibited [5]. Gallagher et al. documented similar findings in the highly metastatic breast cancer cell line MDA-MB-231. In accordance with the findings in MM, lactate efflux was mediated by MCTs and the accessory subunit, CD147. CD147 was highly expressed in MDA-MB-231 cells and its expression was linked to MCT expression. MCT-4 mRNA and protein expression were increased in MDA-MB-231 cells compared to cells derived from normal mammary tissue. MCT-4 co-localized with CD147 in the plasma membrane. CD147 silencing resulted in the loss of MCT4 in the plasma membrane and the accumulation of the transporter in endo-lysosomes. On the other hand, the silencing of MCT-4 impaired the maturation and trafficking of CD147 to the cell surface, resulting in the accumulation of CD147 in the endoplasmic reticulum [49]. Recent studies revealed the involvement of CD147 in cancer cell glycolysis in various malignant tumors including non-small-cell lung cancer, hepatocellular carcinoma, colorectal cancer, prostate cancer, and anaplastic large-cell lymphoma [50,51,52,53,54,55]. These findings strongly suggest that CD147 interacts with MCT-1 and MCT-4 to promote glycolysis in tumor cells, resulting in tumor progression (Figure 2).

## 6. T Cell Differentiation/Proliferation and Glycolysis

Glycolysis is also important for the differentiation, proliferation, and activation of lymphocytes, including T cells, B cells, and natural killer cells. Activated lymphocytes engage in robust growth and rapid proliferation. For these processes, lymphocytes adopt glycolysis [8]. Stimulated CD4^+^ T cells differentiate into effector T cells or inducible regulatory T cells. The differentiation of CD4^+^ T cells into distinct subsets, Th1, Th2, and Th17 cells, requires aerobic glycolysis. Th1, Th2, and Th17 cells express high surface levels of GLUT-1 and are highly glycolytic [56]. Halestrap and Wilson demonstrated the importance of MCT-1 in T cell activation and proliferation. In T cells, energy metabolism is largely glycolytic even under aerobic conditions, and lactic acid efflux from T cells is mediated by MCT-1. MCT-1 is important during the activation and proliferation of resting T cells, which is accompanied by a switch from aerobic to glycolytic metabolism and a remarkable increase in lactate production and export [57]. In line with this, Murray et al. showed that a strong and specific inhibitor targeting MCT-1 functioned as an immunosuppressive drug and prevented T cell proliferation [58]. T cell glycolysis is regulated by CD147 because MCT-1 requires CD147 as an ancillary protein in T cells [57]. Studies on the role of CD147 in T cell biology by Hahn et al. documented that energy metabolism depends on CD147, which is also involved in T cell development, activation, proliferation, migration, invasion, and adhesion. The rapid proliferation and activation of T cells require glycolysis instead of oxidative phosphorylation to respond to their energy demands. Although glycolysis generates energy faster than oxidative phosphorylation, energy production is less efficient and leads to the intracellular accumulation of lactic acid. CD147 plays a role in alleviating lactate efflux for cell stability [59]. In patients with RA, CD147 mRNA expression was elevated in peripheral blood mononuclear cells. Th17 cell differentiation from CD4^+^ T cells is facilitated by CD147 and induces the production of Th17-secreting cytokine, interleukin (IL)-17, and Th17-differentiation-regulating cytokines, IL-6 and IL-1β [60]. These results strongly imply a role for CD147 in the pathophysiology of immune disorders mediated by T cells.

## 7. Psoriasis

Psoriasis is a chronic inflammatory keratotic dermatosis characterized clinically by recurrent episodes of sharply demarcated scaly erythematous plaques and histologically by hyperkeratosis with parakeratosis, Munro microabscess, absence of granular layer, regular elongation of the rete ridges, marked dilatation of blood vessels in the papillary dermis, and dense clusters of inflammatory cells consisting of T cells and dendritic cells in the dermis. Our molecular and cellular understanding of the immunopathogenesis of psoriasis has progressed, and recent studies have revealed that Th17 cells and related signaling pathways play pivotal roles in the development of psoriasis [9,10,61]. First, tumor necrosis factor (TNF)- and inducible nitric oxide synthase (iNOS)-producing dendritic cells (TIP-DC), which reside in the dermis, are activated and produce TNF-α and IL-23. TNF-α is required for the maintenance of the activated state of TIP-DC, while IL-23 promotes the differentiation of naïve CD4^+^ T cells into Th17 cells in humans. Th17 cells produce the effector cytokines IL-17 and IL-22, which activate signal transducer and activator of transcription (STAT) 3 signaling in keratinocytes, resulting in the synthesis of proinflammatory mediators, chemokines, and cytokines like IL-6, IL-8, GM-CSF, CXCL10, and CCL20, which drive the proliferation of epidermal keratinocytes to develop psoriasis lesions [62]. Th17 cells are a distinct T cell subset derived from naïve CD4^+^ helper T cells and the mechanism underlying Th17 cell differentiation is known [63]. The combination of IL-6 and transforming growth factor-β (TGF-β) activates the retinoic acid receptor-related orphan nuclear receptor γt (RORγt), which is the key transcription factor directing the differentiation into Th17 cells [64]. Because the differentiation of naïve CD4^+^ helper T cells into Th17 cells is the fundamental process of psoriasis development [65,66] and requires glycolysis as an energy source, it is possible that CD147 participates in the pathogenesis of psoriasis.

### CD147/Basigin and Psoriasis

Previous studies showed that CD147 is highly expressed in neutrophils in the peripheral blood in the lesional skin of patients with psoriasis [67]. The expression level was significantly correlated with disease severity, as evaluated by the psoriasis area and severity index (PASI) [67]. In this study, the effect of CD147 on neutrophil chemotaxis was presented. An accepted model system for studying neutrophil chemotaxis using all-*trans* retinoic acid (ATRA)-induced differentiated HL-60 promyelocytic leukemia cells was employed. Through treatment with ATRA, HL-60 cells were differentiated into neutrophils, as confirmed by the expression of CD11b, cytoplasm-to-nucleus ratio, and nuclear segmentation accompanied by chromatin condensation. IL-8 and N-formyl-methyonylleucyl-phenylalanine (fMLP), well-known neutrophil chemo-attractants, induced the chemotaxis of ATRA-treated HL-60 cells. The silencing of CD147 significantly decreased the migration of these cells [67]. CD147 is cleaved and released into the peripheral blood as a 22-kDa fragment of the N-terminal extracellular domain [68]. Soluble CD147 has been detected in patients with malignant tumors such as MM and immune disorders including psoriasis [69] and systemic lupus erythematosus (SLE) [70]. In patients with psoriasis, the serum level of soluble CD147 is significantly elevated compared to that in normal controls and correlates with PASI [69]. The effect of soluble CD147 on keratinocyte proliferation was examined. When HaCaT cells, an immortalized non-tumorigenic keratinocyte cell line, were treated with plasma from patients with psoriasis, proliferation was induced compared with that from normal subjects [69].

Regarding T cells in psoriasis, CD147 expression is increased in the CD3^+^ T cells in the dermis of psoriasis lesions. Compared to CD4^+^ RORγt-non-Th17 T cells, its expression was substantially higher in CD4^+^ RORγt^+^ Th17 cells. The degree of MCT-1 expression in RORγt^+^ Th17 cells was upregulated and positively correlated with CD147 expression. These findings indicate that, together with MCT-1, CD147 modulates CD4^+^ T cell differentiation into Th17 cells [11].

The involvement of CD147 in psoriasis development was investigated in wild-type (WT) and CD147-deficient mice. MCT-1 was absent in CD4^+^ T cells from CD147-deficient mice. In response to IL-6 and TGF-β stimulation, which are the inducers of Th17 differentiation, naïve CD4^+^ T cells from the spleen of CD147-deficient mice had a low potential to differentiate into Th17 cells (Figure 3). The topical application of IMQ, a Toll-like receptor-7 agonist, on shaved skin induces psoriasis-like dermatitis. IMQ-induced dermatitis is an accepted psoriasis mouse model. After shaving the dorsal skin of CD147-deficient mice, they were divided into 2 groups; 5% IMQ cream was applied every day for seven days in a row in one group and the other group was treated with ointment base alone. WT mice in the control group were also shaved and divided in the same way. The modified PASI score, which is based on the cumulative disease severity scoring system, was utilized to assess the extent of skin inflammation. Skin erythema, induration, and scaling were categorized into four levels: 0 for none, 1 for mild, 2 for moderate, 3 for severe, and 4 for maximum. A theoretical maximum score of 12 was obtained by adding these scores. Compared to WT mice, the IMQ-induced skin inflammation was noticeably less severe in CD147-deficient mice (Figure 4). The modified PASI scores in WT mice and CD147-deficient mice were 8.75 and 5.2 at day 6, and 8.0 and 4.6 at day 7, respectively. Direct evidence that CD147 plays a significant role in the development of psoriasis was obtained from these studies conducted on mice lacking the *CD147* gene [11].

Various types of cells, including T lymphocytes and epidermal keratinocytes, express CD147. Through the synthesis of IL-22 and the subsequent STAT3 activation in K5-promotor transgenic mice engineered to overexpress CD147 in epidermal keratinocytes, epidermal CD147 contributes to the pathophysiology of psoriasis [71]. Bone marrow chimeric mice devoid of CD147 in the bone marrow hematopoietic cells were created in order to investigate the role of CD147 in immune cells in psoriasis. Ly5.1 recipient mice were subjected to total body irradiation at a dose of 10 Gy in order to generate bone marrow chimera mice. The femurs of CD147-deficient and wild-type (WT) mice were used to harvest bone marrow lumps, which were then transvenously transferred to the recipient mice. CD45.2 and CD45.1 were expressed by leukocytes from C57BL/6J donors and Ly5.1 recipients, respectively, which allows the distinction of the origin of the immune cells in the lesional skin. Compared to WT chimeric mice, the skin lesions induced by IMQ appeared noticeably milder in the CD147-deficient chimeric mice. The modified PASI scores in the WT chimeric mice and CD147-deficient chimeric mice were 5.33 and 1.8 at day 6, and 5.67 and 1.4 at day 7, respectively [11]. These findings demonstrate that the development of psoriasis is caused by CD147 expressed in immune cells.

In the epidermal keratinocytes of psoriatic lesions, metabolism is reprogrammed from oxidative phosphorylation to glycolysis. In addition to the induction of glycolysis, CD147 suppresses the production of carnitine and α-ketoglutaric acid, which are key metabolites in oxidative phosphorylation. The expression of a crucial molecule for carnitine metabolism, γ-butyrobetaine hydroxylase, is also inhibited by CD147 through histone trimethylations of H3K9 in psoriatic keratinocytes [72].

## 8. CD147/Basigin and Other Immune Disorders

The involvement of CD147 in other immune disorders has been previously documented. As mentioned above, CD147 promotes Th17 cell differentiation in patients with RA. The expression of CD147 mRNA was approximately threefold higher in the peripheral blood mononuclear cells (PBMCs) from RA patients than in those from normal subjects. The differentiation of CD4^+^ T cells into Th17 cells was promoted when CD4^+^ T cells were co-cultured with CD14^+^ monocytes in response to lipopolysaccharide (LPS) and anti-CD3 stimulation. The proportion of Th17 cells was significantly higher in RA patients than in normal subjects. Th17-secreting cytokine, IL-17, and Th17-regulating cytokines, IL-6 and IL-1β, were highly induced in the culture supernatants of the CD4^+^/CD14^+^ co-culture. Anti-CD147 antibody reduced the proportion of Th17 cells and inhibited the production of IL-17, IL-6, and IL-1β [60].

Lung interstitial fibrosis is a chronic pulmonary disease characterized by the excessive accumulation of extracellular matrix and often develops as a complication of various autoimmune diseases [73]. In a model of lung interstitial fibrosis in mice treated with bleomycin hydrochloride through the trachea, neutralizing monoclonal antibodies against CD147 (HAb18 mAbs) markedly reduced collagen accumulation and downregulated the proportion of inflammatory M1 macrophages and Th17 cells. In vitro, M1 macrophages induced Th17 differentiation, which was significantly inhibited by treatment with HAb18 mAbs or by silencing CD147 via lentivirus interference in M1 macrophages. These observations indicate that CD147 promotes M1 macrophages and induces Th17 cell differentiation in lung interstitial fibrosis [74].

Multiple sclerosis (MS) is a demyelinating and neurodegenerative autoimmune disorder associated with the migration of activated lymphocytes and macrophages into the central nervous system (CNS). These leukocytes enter the white matter by penetrating the BBB in areas of inflammation called perivascular cuffs and through the CNS–meningeal barrier. The accumulation of activated leukocytes in the lesion causes the loss of BBB integrity, leading to the transmigration of leukocytes into the brain parenchyma. A metabolic switch is seen in T cells and macrophages in the lesion; they heavily depend on glycolysis as their energy source. In the experimental autoimmune encephalomyelitis (EAE) model of MS, macrophages within the perivascular cuffs of postcapillary venules were highly glycolytic, as shown by the strong expression of lactate dehydrogenase (LDHA), which converts pyruvate to lactate. These macrophages expressed a prominent level of MCT-4, which engages the secretion of lactate from the cytoplasm of glycolytic cells. The silencing of LDHA or MCT-4 by each siRNA resulted in decreased lactate secretion and macrophage transmigration, indicating the functional relevance of glycolysis in the pathogenesis of EAE. In EAE, CD147 binds to MCT-4 and regulates its expression on the cell membrane of macrophages, which was confirmed by co-immunoprecipitation. Glycolysis and glycolytic capacity were significantly abrogated by CD147-knockdown by siRNA, as measured by the extracellular acidification rate (ECAR). These results demonstrate the importance of CD147, in association with MCT-4, in governing macrophage glycolysis and its migration in inflammatory perivascular cuffs of EAE. The relevance of MS was confirmed by the strong expression of CD147, MCT-4, and LDHA in the perivascular macrophages in the brains of patients with MS [75].

## 9. Conclusions and Future Directions

CD147 functions as a chaperone for various membrane proteins including CyPA, integrins, P-gp, and MCTs, among others, and it supports their plasma membrane expression. In this review, we focused on MCTs. In association with MCT-1 and MCT-4, CD147 regulates glycolysis, an enzymatic metabolic system that generates energy that is stored as ATP. Glycolysis is required by cancer cells for their proliferation, invasiveness, and VEGF production and by T cells for their differentiation, proliferation, and activation. CD147 contributes to the development of malignant tumors and Th17-cell-mediated immune disorders including psoriasis. Therefore, CD147 is a promising therapeutic target for patients with these disorders. Inhibitors of CD147, MCTs, or CD147/MCT complex would be effective. The other option is anti-CD147 antibodies, which block the function of CD147. For skin diseases such as psoriasis, topical application could be adopted. Compounds are able to be delivered to the skin as an ointment with a petrolatum base and 5% stearyl alcohol. For diseases of the internal organs, an appropriate drug delivery system is needed. The usefulness of nanoparticles was reported. The therapeutic effect of nanoparticles carrying the anti-CD147 antibody was demonstrated in lung cancer and hepatocellular carcinoma [76,77,78]. The anti-CD147 antibody was bound to dextran nanoparticles in the trial for lung cancer [76], conjugated to liposomes for hepatoma [77], or to phosphoester polymeric nanoparticles for hepatocellular carcinoma [78]. The possible delivery systems for compounds or antibodies targeting CD147 that will be applied in the future are summarized in Table 1.

## Figures and Tables

**Figure 1 ijms-24-17344-f001:**
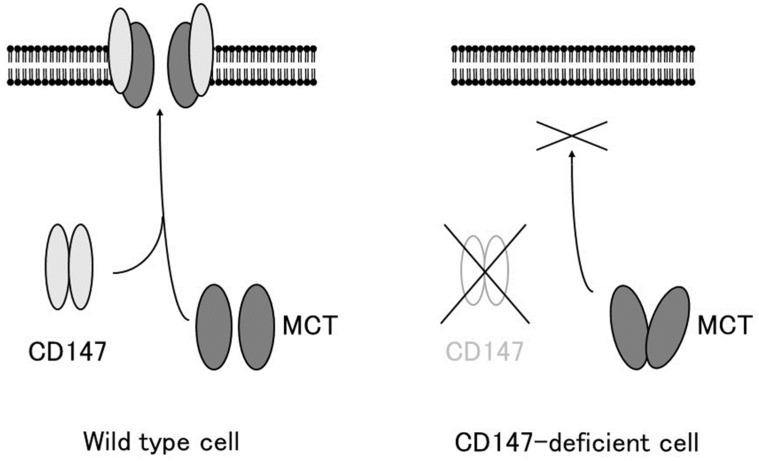
CD147 associates with MCT-1 and MCT-4 and facilitates their expression in the plasma membrane. In the absence of CD147, MCTs accumulate in the cytosol and are not expressed on the plasma membrane (modified from [7]).

**Figure 2 ijms-24-17344-f002:**
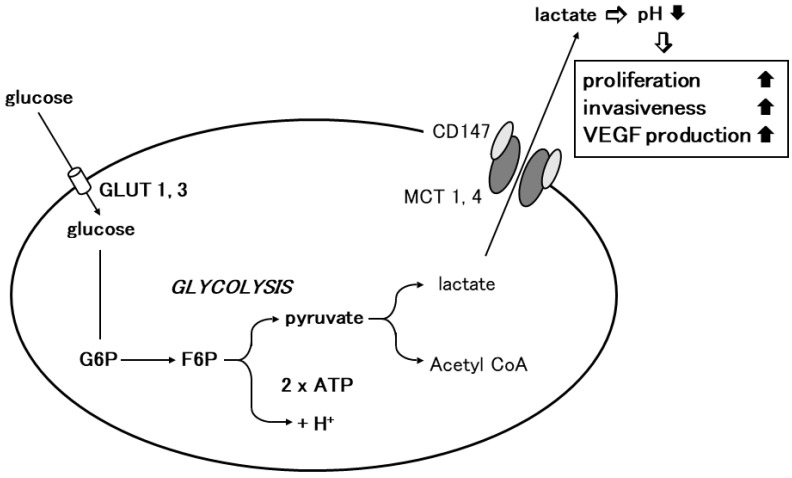
Involvement of CD147 in tumor cell glycolysis. Glycolysis is the enzymatic conversion of glucose to pyruvate to produce ATP. Cancer cells depend on glycolysis for energy aerobic conditions. Cancer cells take up excess glucose through GLUT-1 or GLUT-3, which is enzymatically converted to ATP to provide energy for pathophysiological processes such as cellular growth and proliferation. Through MCT-1 or MCT-4, pyruvate is further transformed into lactic acid during aerobic glycolysis in cancer cells, which is then released from the cytoplasm into the surrounding extracellular milieu. Homodimerized CD147 associates with two monomers of MCT1 or 4 and regulates lactate transport (modified from [7]).

**Figure 3 ijms-24-17344-f003:**
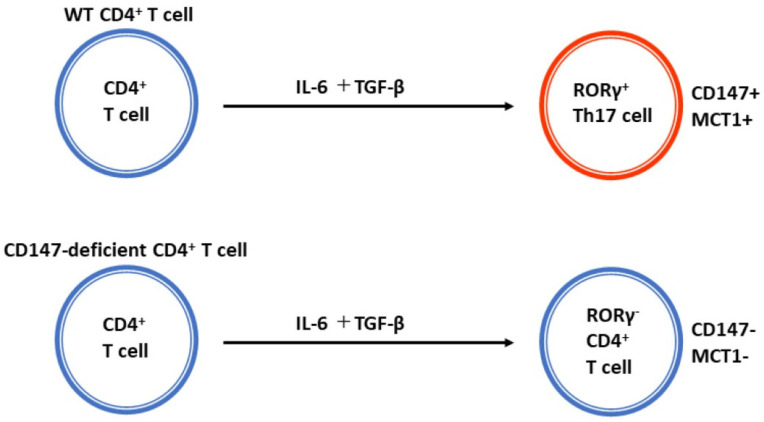
Impact of CD147 deficiency in vitro on MCT-1 and RORγt expression in CD4^+^ T cells. Spleens of CD147-deficient and wild-type (WT) mice were used to isolate naïve CD4^+^ T cells, which were then cultivated on chamber slides and stimulated with IL-6; TGF-βRORγt expression was observed in WT CD4^+^ T cells, whereas CD147-deficient CD4^+^ T cells exhibited a noticeably lower expression. WT CD4^+^ T cell plasma membranes expressed MCT-1, whereas CD147-deficient CD4^+^ T cells did not express it (modified from [11]).

**Figure 4 ijms-24-17344-f004:**
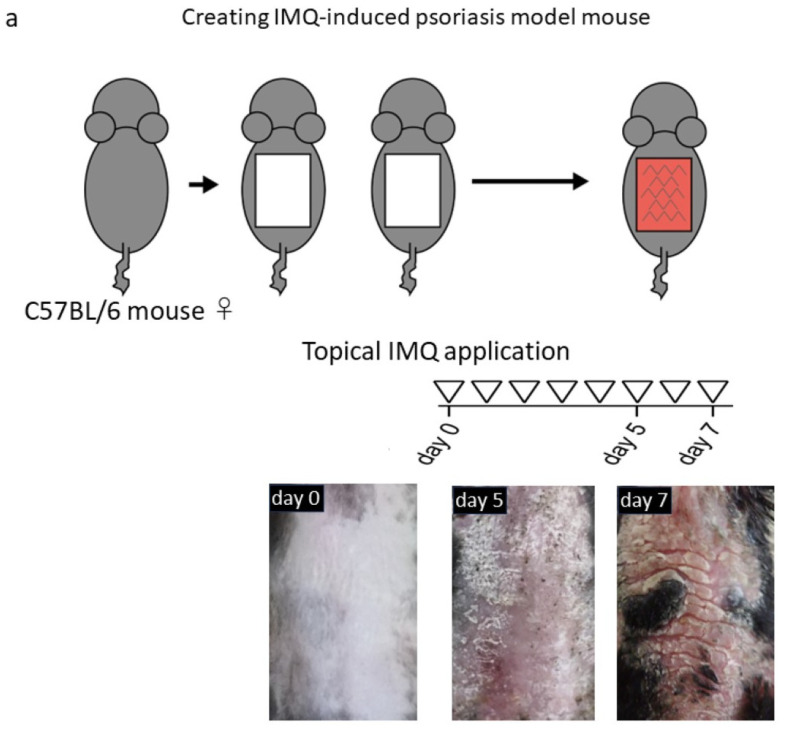
Impact of CD147 deficiency on the development of dermatitis induced by imiquimod (IMQ). (**a**) Topical application of IMQ for seven consecutive days induced psoriasis-like dermatitis. (**b**) Modified PASI score was significantly lower in CD147-deficient mice than in WT mice at day 6 and day 7.

**Table 1 ijms-24-17344-t001:** Possible delivery system of drugs targeting CD147.

		Skin Diseases	Internal Diseases
CD147/MCT inhibitor	Topical application	applicable	
	Nanoparticle carrier	applicable	applicable
CD147 Ab *	Topical application	applicable	
	Nanoparticle carrier	applicable	applicable

* Ab: antibody.

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
