# Peer review of "CD147/Basigin Is Involved in the Development of Malignant Tumors and T-Cell-Mediated Immunological Disorders via Regulation of Glycolysis"

_ijms, 2023, doi:10.3390/ijms242417344_

Round 1

Reviewer 1 Report

Comments and Suggestions for Authors

The authors wrote the paper "CD147/Basigin is involved in the development of malignant tumors and T cell mediated immunological disorders via regulation of glycolysis". While the tumor and immunological implications of CD147/Basigin are interesting, I would like to see the paper fleshed out. 

1. Figure 1 is a difficult picture to understand. It needs to be refined. 

2. The authors need more clarification on the results of the latest paper on CD147 and glycolysis. 

3. The author needs to rethink whether the content of the Psoriasis paragraph is necessary in this paper. 

4. need statistical figures for Figures 3 and 4. 

Author Response

I appreciate the detailed review of the manuscript. The comments of the reviewer have been helpful in my revision of the manuscript.  I have carefully considered and addressed each of the comments, as described below, in the revised manuscript.

1. In accordance with the reviewer’s comment, I modified Figure 1 to make it easily understandable.

2. I added the description of the results of the latest articles on CD147 and glycolysis (lines 344-349).

3. I understand what the reviewer means and reconsider carefully about the “Psoriasis paragraph”. With all due respect to the reviewer, I believe this paragraph is necessary in order to emphasize that the differentiation of naïve CD4 T cells into Th17 cells is the fundamental process in the pathogenesis of psoriasis.

4. The reviewer is right to point out that statistical figures are needed for Figures 3 and 4. According to the Academic Editor’s instruction, Figure 3 was replaced by the schematic figure. Figure 4 was also replaced and the statistical figure was added.

Reviewer 2 Report

Comments and Suggestions for Authors

This manuscript deals with the role of CD147 in cancer, and immune disorders related to the glycolysis.

It appears that this manuscript has been submitted to an issue on skin diseases. Actually, the topic is focus also on immune disorders other than skin diseases. Regarding the cancer related section, the CD147 is considered mainly in melanoma. 

I would erase the information on other immune disorders, and I would focus on skin diseases exclusively. Otherwise, all cancers should be considered.

A table with the mean to target CD147 could help the reader to understand better the therapies that will be applied in the future.

In this context, it would be of help to cite some papers on nanoparticles (putting the words Cd147 and nanoparticles some papers can be considered in PUBMED). 

Comments on the Quality of English Language

English language is good.

Author Response

I am grateful for the detailed review of the manuscript. The comments of the reviewer have been helpful in my revision of the manuscript.  I have carefully considered and addressed each of the comments, as described below, in the revised manuscript.

1. I appreciate the reviewer’s comment to focus on skin diseases exclusively, but yet I believe it is worth describing the other diseases. Because it is important to show that the role of CD147 in glycolysis is common in various diseases.

2. I added the information about other malignant tumors (lines 203-206).

3. Thank you for the meaningful suggestion. I provided the table with the means to target CD147. Accordingly, the description of possible therapeutic approaches was added to the “Conclusion and future directions” section (lines 403-410).

Round 2

Reviewer 1 Report

Comments and Suggestions for Authors

1. The review comments are written in a way that makes it difficult to understand the answer. 

2. Figures are a complement, but are not a substitute for resolution and clarity.

3. Table 1 can be explained in a little more detail.

Author Response

I appreciate the detailed review of the manuscript. I have carefully considered and addressed each of the comments, as described below, in the revised manuscript.

  1. I apologize for my reply which was written in a way that makes it difficult to understand the answer.  Here, I reply again  to the comments of Round 1.

  1. I refined Figure 1 in accordance with the reviewer’s comment.
  2. I added the results of the latest paper on CD147 and glycolysis (line 344-349. Reference 72.)
  3. We thought that the “psoriasis paragraph” is necessary. Because this section describes that the differentiation of naïve CD4+ T cells into Th17 cells is the fundamental process in the development of psoriasis lesions.
  4. After submitting the original version, I received the suggestion from the Academic Editor. The suggestion is as follows. "I have not asked for an improvement in the resolution of the figures. Replace Figures 3 and 4 with entirely new photos that differ from those presented in Ref.11." The Editorial Office advised that "academic editor expects you to have other photos from laboratory showing the same content as in Ref.11.”

    According to this suggestion, I replaced the Figures 3 with schematic      presentation which shows the same content as in Ref. 11. Figure 4 was also replaced and added the statistical figure.  

  1. I agree with the reviewer that figures are complement but are not a substitute for resolution and clarity. Actually, I resubmitted the figures with high resolution after the first submission, then I received the suggestion as mentioned above. According to this suggestion, I replaced the Figures 3 with schematic presentation which shows the same content as in Ref. 11. Figure 4 was also replaced and added the statistical figure. 

  1. I added a little more detailed explanation of Table 1 to the “Conclusion and future directions” section (lines 404-415).

Round 3

Reviewer 1 Report

Comments and Suggestions for Authors

I think the manuscript has been revised well.